# Menopausal hormone therapy and women's health: An umbrella review

**Guo-Qiang Zhang** [1]*, **Jin-Liang Chen**[2☯], **Ying Luo**[3☯], **Maya B. Mathur** [4☯], **Panagiotis Anagnostis** [5], **Ulugbek Nurmatov**[6], **Madar Talibov** [7], **Jing Zhang**[8], **Catherine M. Hawrylowicz** [9], **Mary Ann Lumsden**[10], **Hilary Critchley**[11], **Aziz Sheikh** [12], **Bo Lundbäck**[1], **Cecilia Lässer** [1], **Hannu Kankaanranta** [1,13,14], **Siew Hwa Lee** [15], **Bright I. Nwaru** [1,12,16]

1 Krefting Research Centre, Institute of Medicine, Sahlgrenska Academy, University of Gothenburg, Gothenburg, Sweden, 2 Department of Geriatrics, First Affiliated Hospital of Chongqing Medical University, Chongqing, China, 3 Department of Gastroenterology, Children's Hospital of Chongqing Medical University, National Clinical Research Center for Child Health and Disorders, Ministry of Education Key Laboratory of Child Development and Disorders, China International Science and Technology Cooperation Base of Child Development and Critical Disorders, Chongqing, China, 4 Quantitative Sciences Unit, Stanford University, Palo Alto, California, United States of America, 5 Unit of Reproductive Endocrinology, 1st Department of Obstetrics and Gynecology, Medical School, Aristotle University of Thessaloniki, Thessaloniki, Greece, 6 Division of Population Medicine, School of Medicine, Cardiff University, Cardiff, United Kingdom, 7 UMR1086 INSERM–Unité de Recherche Interdisciplinaire pour la Prévention et le Traitement des Cancers (ANTICIPE), Centre de Lutte Contre le Cancer François Baclesse, Caen, France, 8 Department of Intensive Care Unit, Chongqing General Hospital, University of Chinese Academy of Sciences, Chongqing, China, 9 MRC and Asthma UK Centre in Allergic Mechanisms of Asthma, King's College London, Guy's Hospital, London, United Kingdom, 10 Section of Reproductive and Maternal Medicine, Glasgow Royal Infirmary, School of Medicine, University of Glasgow, Glasgow, United Kingdom, 11 MRC Centre for Reproductive Health, Queen's Medical Research Institute, University of Edinburgh, Edinburgh, United Kingdom, 12 Asthma UK Centre for Applied Research, Centre for Medical Informatics, Usher Institute, University of Edinburgh, Edinburgh, United Kingdom, 13 Department of Respiratory Medicine, Seinäjoki Central Hospital, Seinäjoki, Finland, 14 Faculty of Medicine and Health Technology, University of Tampere, Tampere, Finland, 15 School of Nursing, Midwifery and Paramedic Practice, Robert Gordon University, Aberdeen, United Kingdom, 16 Wallenberg Centre for Molecular and Translational Medicine, University of Gothenburg, Gothenburg, Sweden

☯ These authors contributed equally to this work.
* guo-qiang.zhang@gu.se

**Data Availability Statement:** The statistical analysis protocol and R scripts are respectively available in S3 Text and S4 and S5 Texts. The datasets are publicly available at the Open Science

## Abstract

### Background

There remains uncertainty about the impact of menopausal hormone therapy (MHT) on women's health. A systematic, comprehensive assessment of the effects on multiple outcomes is lacking. We conducted an umbrella review to comprehensively summarize evidence on the benefits and harms of MHT across diverse health outcomes.

### Methods and findings

We searched MEDLINE, EMBASE, and 10 other databases from inception to November 26, 2017, updated on December 17, 2020, to identify systematic reviews or meta-analyses of randomized controlled trials (RCTs) and observational studies investigating effects of MHT, including estrogen-alone therapy (ET) and estrogen plus progestin therapy (EPT), in perimenopausal or postmenopausal women in all countries and settings. All health

Framework (https://osf.io/dsy37/; DOI: 10.17605/OSF.IO/DSY37).

**Funding:** GQZ was supported by the VBG Group Herman Krefting Foundation on Asthma and Allergy, and Sahlgrenska Academy, University of Gothenburg, Sweden. MBM was supported in part by the Franklin E. Fetzer Memorial Trust (award R2020-16). BIN was supported by the VBG Group Herman Krefting Foundation on Asthma and Allergy, Knut and Alice Wallenberg Foundation, and the Wallenberg Centre for Molecular and Translational Medicine, University of Gothenburg, Sweden. The funders had no role in study design, data collection and analysis, decision to publish, or preparation of the manuscript.

**Competing interests:** I have read the journal's policy and the authors of this manuscript have the following competing interests: HC reports no conflicts specifically related to menopausal hormone therapy, and in the more general area of women's health the following are declared: Grant/Research Support in AUB treatment: Bayer AG; TAP Pharmaceuticals, Preglem/Gedeon-Richter: PEARL IV (with no personal remuneration); Speaker bureau on AUB Treatment: Bayer AG, Preglem/Gedeon-Richter, Myovant (with no personal remuneration); Consultancy/Advisory Board on AUB Treatment: Bayer AG, Preglem/Gedeon-Richter, Vifor Pharma, Myovant (with no personal remuneration); Royalties from UpToDate: for an article on abnormal uterine bleeding (personal remuneration received). AS is an Academic Editor on PLOS Medicine's editorial board. CL has co-developed extracellular vesicle-associated patents for putative clinical utilization and owns equity in Exocure Bioscience Inc. HK reports personal fees and non-financial support from AstraZeneca, personal fees from Chiesi Pharma AB, personal fees and non-financial support from Boehringer-Ingelheim, personal fees from Novartis, personal fees from Mundipharma, personal fees and non-financial support from Orion Pharma, personal fees from SanofiGenzyme, personal fees from GlaxoSmithKline, outside the submitted work. All other authors have declared that no competing interests exist.

**Abbreviations:** CHD, coronary heart disease; CI, confidence interval; EPT, estrogen plus progestin therapy; ET, estrogen-alone therapy; LDL-C, low-density lipoprotein cholesterol; MHT, menopausal hormone therapy; PI, prediction interval; PI(E)COS, Population, Intervention or Exposure, Comparator, Outcome, Study design; RCT, randomized controlled trial; RR, risk ratio; WHI, Women's Health Initiative.

outcomes in previous systematic reviews were included, including menopausal symptoms, surrogate endpoints, biomarkers, various morbidity outcomes, and mortality. Two investigators independently extracted data and assessed methodological quality of systematic reviews using the updated 16-item AMSTAR 2 instrument. Random-effects robust variance estimation was used to combine effect estimates, and 95% prediction intervals (PIs) were calculated whenever possible. We used the term MHT to encompass ET and EPT, and results are presented for MHT for each outcome, unless otherwise indicated. Sixty systematic reviews were included, involving 102 meta-analyses of RCTs and 38 of observational studies, with 102 unique outcomes. The overall quality of included systematic reviews was moderate to poor. In meta-analyses of RCTs, MHT was beneficial for vasomotor symptoms (frequency: 9 trials, 1,104 women, risk ratio [RR] 0.43, 95% CI 0.33 to 0.57, $p < 0.001$; severity: 7 trials, 503 women, RR 0.29, 95% CI 0.17 to 0.50, $p = 0.002$) and all fracture (30 trials, 43,188 women, RR 0.72, 95% CI 0.62 to 0.84, $p = 0.002$, 95% PI 0.58 to 0.87), as well as vaginal atrophy (intravaginal ET), sexual function, vertebral and nonvertebral fracture, diabetes mellitus, cardiovascular mortality (ET), and colorectal cancer (EPT), but harmful for stroke (17 trials, 37,272 women, RR 1.17, 95% CI 1.05 to 1.29, $p = 0.027$) and venous thromboembolism (23 trials, 42,292 women, RR 1.60, 95% CI 0.99 to 2.58, $p = 0.052$, 95% PI 1.03 to 2.99), as well as cardiovascular disease incidence and recurrence, cerebrovascular disease, nonfatal stroke, deep vein thrombosis, gallbladder disease requiring surgery, and lung cancer mortality (EPT). In meta-analyses of observational studies, MHT was associated with decreased risks of cataract, glioma, and esophageal, gastric, and colorectal cancer, but increased risks of pulmonary embolism, cholelithiasis, asthma, meningioma, and thyroid, breast, and ovarian cancer. ET and EPT had opposite effects for endometrial cancer, endometrial hyperplasia, and Alzheimer disease. The major limitations include the inability to address the varying effects of MHT by type, dose, formulation, duration of use, route of administration, and age of initiation and to take into account the quality of individual studies included in the systematic reviews. The study protocol is publicly available on PROSPERO (CRD42017083412).

## Conclusions

MHT has a complex balance of benefits and harms on multiple health outcomes. Some effects differ qualitatively between ET and EPT. The quality of available evidence is only moderate to poor.

## Author summary

### Why was this study done?

- By 2050, it is estimated that worldwide more than 1.6 billion women will have reached menopause or be postmenopausal, up from 1 billion in 2020.

- Up to 75% of menopausal women are affected by bothersome menopausal symptoms, such as hot flashes and night sweats.

- Menopausal hormone therapy (MHT) is the most effective treatment for alleviating menopausal symptoms, but its effects on numerous health outcomes remain uncertain.

## What did the researchers do and find?

- We included 60 published systematic reviews of MHT use in menopausal women, involving 102 meta-analyses of randomized controlled trials and 38 of observational studies, and synthesized the evidence on 102 health outcomes.

- Overall, MHT had a complex balance of benefits and harms; for example, beyond alleviation of menopausal symptoms, it was associated with decreased risks of bone fracture, diabetes mellitus, and esophageal, gastric, and colorectal cancer, but increased risks of stroke, venous thromboembolism, gallbladder disease, and breast and ovarian cancer.

- The available clinical data in support of MHT reducing the risk of coronary heart disease and all-cause mortality in women aged <60 years or within 10 years from menopause (known as the "timing hypothesis") were only suggestive.

- The overall quality of included systematic reviews was moderate to poor.

## What do these findings mean?

- This overview of the evidence landscape could help guideline developers and decision-makers better appreciate the trade-offs between the benefits and harms associated with MHT use in menopausal women.

- More data are needed to evaluate the timing hypothesis for coronary heart disease and all-cause mortality.

- Clinicians should evaluate the scientific strength of systematic reviews prior to considering applying their results in clinical practice.

## Introduction

Longevity is increasing worldwide for women. By 2050, the world's women aged 50 years and older are projected to total 1.6 billion, up from 1 billion in 2020 [1]. Natural menopause occurs at a mean age of 49 years [2]. Vasomotor symptoms, including hot flashes and night sweats, are the hallmark symptoms of menopause, affecting approximately 75% of perimenopausal women, and may persist for a decade or longer [3]. In addition, up to 84% of postmenopausal women experience genitourinary symptoms, such as vulvovaginal atrophy and incontinence [4]. The burden of menopausal symptoms can considerably affect the personal, social, and work lives of women [3]. Menopausal hormone therapy (MHT) is the most effective treatment for managing vasomotor and genitourinary symptoms [5,6]. In high-income countries, there were about 600 million woman-years of MHT use in the period 1970–2019, and about 12 million users in the 2010s, of whom 6 million users were in the US and UK alone [7,8].

For several decades, the possible health effects of MHT, beyond alleviation of menopausal symptoms, have been debated. Nevertheless, a systematic, comprehensive assessment in this regard is lacking. More recently, leading medical societies provided clinical practice guidelines for the use of MHT in women [5,6,9–11]. The guidelines lacked consistency regarding some outcomes, such as coronary heart disease (CHD) and all-cause mortality [5,6,9–11]. Often, these guidelines incorporated systematic reviews and meta-analyses as key evidence support for the recommendations, but with little attention to their quality or scientific validity [5,9]. Numerous empirical evaluations have found that poor quality and major flaws impede many published systematic reviews in diverse disciplines [12–18]. In light of the uncertainty surrounding the effects of MHT on numerous health outcomes in women, it is important that the quality of systematic reviews contributing to current guidelines and recommendations be critically appraised in order to provide the highest level and most reliable basis for recommendations in clinical practice. Consequently, we performed an outcome-wide umbrella review to summarize the evidence across existing systematic reviews and meta-analyses on the effects of MHT in perimenopausal and postmenopausal women [19]. We sought to provide a comprehensive overview of the benefits and harms of MHT use, evaluate the validity of effects reported across systematic reviews, assess potential biases in the literature, and examine the credibility of the methods of existing systematic reviews.

## Methods

An umbrella review is a review of published systematic reviews, with or without meta-analyses, related to a given topic or question [20–22]. It systematically integrates evidence from multiple systematic reviews to present a comprehensive view of the evidence landscape, readily providing guideline developers and decision-makers with the currently available highest level of evidence relevant to the question posed [21,22]. The protocol for the current umbrella review was registered in the International Prospective Register of Systematic Reviews (PROSPERO) (CRD42017083412). This study was reported as per the Preferred Reporting Items for Systematic Reviews and Meta-Analyses (PRISMA) guideline [23] (S1 PRISMA Checklist). Ethical approval was not required for this study.

### Literature search and eligibility criteria

We searched MEDLINE, EMBASE, Cochrane Database of Systematic Reviews, Database of Abstracts of Reviews of Effects, ISI Web of Science, CINAHL, Google Scholar, Allied and Complementary Medicine Database, Global Health, PsycINFO, CAB International, and WHO Global Health Library from inception to November 26, 2017. No language restriction was applied. Table A in S1 Text presents the detailed search strategies for each electronic database. Two investigators independently screened the titles and/or abstracts and reviewed full-text articles for eligibility. Any discrepancies were resolved by discussion. References of the included articles were also manually checked to identify additional eligible articles.

Articles were selected for inclusion if they met the following Population, Intervention or Exposure, Comparator, Outcome, Study design (PI[E]COS) criteria: (1) population: perimenopausal or postmenopausal women of any ethnicity in any country or setting; (2) intervention or exposure: any type of MHT, including estrogen-alone therapy (ET) and estrogen plus progestin therapy (EPT), at any dose, duration, and route of administration; (3) comparator: placebo or no treatment; (4) outcome: any health outcome or indicator, including menopausal symptoms; and (5) study design: systematic review, with or without meta-analysis, of randomized controlled trials (RCTs) or observational epidemiological studies (cohort and case–control design). We excluded reviews without a systematic search, systematic reviews that

examined the effects of progestin or selective estrogen receptor modulators (e.g., raloxifene), systematic reviews that compared different types of MHT, and individual participant meta-analyses. If more than 1 systematic review existed on the same scientific question (those with the same PI[E]COS elements), we included the most recent and largest review. In some cases where the largest meta-analysis was not the most recent, we retained the largest and updated it by including the nonoverlapping studies from the most recent review. If the same outcome was investigated in systematic reviews of RCTs and of observational studies or in systematic reviews with different population or intervention/exposure characteristics, all reviews were included for that outcome in order to get a complete picture of the benefits or harms associated with MHT. We categorized outcomes according to the International Classification of Diseases–11th Revision (ICD-11) [24]. We performed an updated search on December 17, 2020, for studies published since November 26, 2017, but the literature obtained did not change the conclusions reached based on the originally included studies. A summary of the literature obtained from the updated search is given in S2 Text.

## Data extraction and quality assessment

Two investigators independently extracted data by outcome using a customized data extraction form. For each systematic review, we extracted title of the review, citation details (e.g., author list, journal, year of publication), country, PI(E)COS elements, number of included studies, and meta-analysis method (where applicable). For each individual study included in each meta-analysis, we extracted first author, year of publication, country, study design, phase of prevention, age of participants, menopausal status, type of MHT, route of administration, length of follow-up (where applicable), outcome examined, number of events for binary outcomes or means and standard deviations for continuous outcomes and total number of participants in intervention and control groups in RCTs, number of cases and controls in case–control studies or total population in cohort studies, type of effect estimate (mean difference, standardized mean difference, risk ratio [RR], odds ratio, incidence rate ratio, or hazard ratio), and effect estimate with 95% confidence interval (CI). For systematic reviews without meta-analysis, we abstracted only key findings or conclusions.

Given the issues (e.g., incomplete data, poor data quality) in a sample of included meta-analyses [25–28] as well as the high prevalence of data extraction errors from empirical evaluations [29], we devoted extensive efforts to obtaining and validating the data for the individual studies in each meta-analysis. We developed 2 separate protocols for extracting data from included systematic reviews of RCTs and of observational studies based on the assessment of 5 included reviews [25–28,30] (Figs A and B in S1 Text). In brief, for meta-analyses of RCTs that provided insufficient or inadequate data for individual studies, the full texts of these studies were retrieved, from which data were extracted. In addition, outcome data from the Women's Health Initiative (WHI) trials were updated based on the most recent publications [31–34]. For meta-analyses of observational studies, we extracted all data from individual studies regardless of the availability of data in the meta-analyses, and then compared these data with the data in the meta-analyses. We verified the inclusion of individual studies in each meta-analysis by checking the eligibility of each study against the PI(E)COS criteria of the meta-analysis, and studies were further excluded if found not to meet the criteria. In a few cases (e.g., no references provided), the review authors were contacted to request additional information.

Two investigators independently assessed the methodological quality of included systematic reviews using the updated 16-item AMSTAR 2 (A MeaSurement Tool to Assess systematic Reviews) instrument [35]. Quality appraisal of individual studies is beyond the scope of an

umbrella review. Any discrepancies during data extraction and quality assessment were resolved by discussion or arbitration by a third investigator.

## Data synthesis and analysis

We developed a priori an analysis protocol including justification of the statistical methods applied to this umbrella review (S3 Text). In brief, for each meta-analysis we calculated the summary average effect and its 95% CI using the random-effects robust variance estimation method [36]. We quantified the extent of heterogeneity by estimating the between-study standard deviation [37,38]. To further account for heterogeneity, we estimated the predictive distribution for the true effect in a new study [39–42]. The 95% prediction interval (PI) estimates the middle 95% area of the predictive distribution, reflecting the variation in the true effects across study settings, and predicts with 95% confidence the true effect in a new study that is similar to the studies in the meta-analysis [40,43]. Based on simulation results [40,42], the 95% PI was calculated only in meta-analyses of ≥10 studies. Next, we examined small-study effects with a random-effects Egger regression [44,45], which assesses whether there is an association between treatment effect size and its standard error. To assess publication bias, we used the Vevea and Hedges selection model [46] and the S-value [47], which represents the severity of publication bias that would hypothetically be required to shift the point estimate to the null. Lastly, for meta-analyses of observational studies, we assessed the robustness of meta-analysis results to potential residual confounding using the E-value [48,49] and its equivalents for meta-analyses [42,50].

The primary analyses focused on the average effects of any type of hormone therapy (ET or EPT). A subgroup analysis by MHT type was conducted to assess whether the effect varied qualitatively between ET and EPT. A qualitative difference means that the effects of ET and EPT do not point in the same direction [51,52]. We use the term MHT to encompass ET and EPT, and results are presented for MHT, unless a qualitative difference was indicated between ET and EPT, in which case results are presented separately for them. We grouped the data according to whether the intervention or exposure was primary or secondary prevention. We defined primary prevention as reducing the risk of occurrence of a disease among individuals who do not have that disease at the beginning of follow-up, and secondary prevention as reducing the severity or progression of a disease among individuals who already have that disease or the risk of recurrence among individuals who have a history of that disease. The analyses were conducted separately for RCTs and observational studies. For observational studies, the analyses were stratified by recency of MHT use (ever, current, or past). Finally, based on the amount and the consistency of the body of evidence, we graded the evidence from meta-analyses as consistent, highly suggestive, suggestive, controversial, or insufficient, following the criteria described in Tables 1 and 2. All estimates were converted to the RR scale [48,53–55], and results are presented on that scale, except where otherwise noted. All statistical analyses were performed using R software (version 4.0.1) [56]. The R scripts are available in S4 and S5 Texts, and the datasets are available at Open Science Framework (https://osf.io/dsy37/).

## Results

### Characteristics of included studies

In total, we identified 10,550 records, scrutinized 160 full-text articles, and ultimately included 60 articles (29 systematic reviews of RCTs [25,26,30,57–82], 27 of observational studies [27,28,83–107], and 4 of both RCTs and observational studies [108–111]) (Fig 1). The studies were published between 1995 and 2017. The 33 systematic reviews of RCTs reported 102 meta-analyses (1 systematic review without meta-analysis), with 81 unique outcomes; the 31

**Table 1. Summary of evidence grading for meta-analyses of randomized controlled trials on menopausal hormone therapy and incidence of diseases and other health outcomes.**

| Evidence | Criteria | Meta-analyses of randomized controlled trials[a] | |
|---|---|---|---|
| | | Outcomes with beneficial effects | Outcomes with harmful effects |
| Consistent | 95% CI of the mean effect excludes null value with no heterogeneity, or predictive distribution[b] contains an extreme proportion (>90%) of true effects in the direction of the mean effect | All fracture, vertebral fracture, nonvertebral fracture, colorectal cancer (EPT), cardiovascular mortality (ET) | Cardiovascular disease, cerebrovascular disease, stroke, nonfatal stroke, venous thromboembolism, deep vein thrombosis, gallbladder disease requiring surgery, endometrial hyperplasia (ET), lung cancer mortality (EPT) |
| Highly suggestive | 95% CI of the mean effect excludes null value, with heterogeneity present but predictive distribution not estimable[c], or predictive distribution contains a substantial proportion (70%–90%) of true effects in the direction of the mean effect | Vasomotor symptoms[d], vasomotor symptom severity[d], vaginal atrophy (intravaginal ET)[d], sexual function, urinary incontinence[d], diabetes mellitus | Cardiovascular disease recurrence |
| Suggestive | 95% CI of the mean effect includes null value, predictive distribution not estimable[c], and 95% CI of the most precise study[e] excludes null value | Breast cancer (ET), endometrial cancer (EPT), hip fracture (EPT), sleep quality (EPT), skeletal muscle strength (EPT), recurrent urinary tract infection (ET)[d] | Breast cancer (EPT), dementia (EPT), pulmonary embolism (EPT), irregular vaginal bleeding (ET) |
| Controversial | Predictive distribution contains a non-negligible proportion (>30%) of true effects in both the same and the opposite direction of the mean effect | None | None |
| Insufficient | Insufficient evidence to draw conclusions | All-cause mortality, all cancer incidence and mortality, lung cancer incidence and mortality (ET), breast cancer recurrence and mortality, ovarian cancer incidence and overall survival, endometrial cancer (ET), colorectal cancer incidence (ET) and mortality, cardiovascular mortality (EPT), cerebrovascular disease recurrence, stroke recurrence, fatal stroke incidence and recurrence, nonfatal stroke recurrence, transient ischemic attack incidence and recurrence, coronary heart disease incidence and recurrence and mortality, myocardial infarction incidence and recurrence, fatal and nonfatal myocardial infarction incidence and recurrence, angina pectoris incidence and recurrence, cardiac death, coronary revascularization, venous thromboembolism recurrence, deep vein thrombosis recurrence, pulmonary embolism incidence (ET) and recurrence, dementia (ET), Alzheimer disease, cognitive function (in healthy women and in women with dementia), hip fracture (ET), endometrial hyperplasia (EPT), irregular vaginal bleeding (EPT), sleep quality (ET), occurrence and recurrence of depressive symptoms | |

Small-study effects existed for all fracture, sexual function, urinary incontinence, and deep vein thrombosis; the Egger regression test was used to examine whether smaller studies tended to show more pronounced effects than larger studies; it was applied only in meta-analyses of ≥10 studies; more information is available in S3 Text. Meta-analysis results were robust to severe or extreme publication bias for all fracture, vasomotor symptoms, stroke, nonfatal stroke, venous thromboembolism, gallbladder disease requiring surgery, and endometrial hyperplasia (ET); "robust to severe or extreme publication bias" means that the meta-analysis results cannot be explained away by hypothetical publication bias that greatly exceeds empirical estimates of publication bias severity in medicine (i.e., hypothetical publication bias in which statistically significant positive effects are 4-fold more likely to be published and meta-analyzed than nonsignificant or negative effects); more information is available in S3 Text. CI, confidence interval; EPT, estrogen plus progestin therapy; ET, estrogen-alone therapy.

[a]The average effects of any menopausal hormone therapy (ET or EPT) in perimenopausal or postmenopausal women. When subgroup analysis by type of hormone therapy indicated a qualitative difference or statistically significant results were found for only 1 type of hormone therapy, results are presented separately for them. The effects refer to outcome incidence, unless otherwise indicated.

[b]The predictive distribution describes how the true effect sizes across studies are distributed around the summary average effect. Predictive distribution was estimated only in meta-analyses of ≥10 studies. More information is available in S3 Text.

[c]Due to a small number of studies (<10) being included in the meta-analysis.

[d]In women who already have the outcome of interest.

[e]The study with the smallest standard error in each meta-analysis.

systematic reviews of observational studies reported 38 meta-analyses (2 systematic reviews without meta-analysis), with 40 unique outcomes; 19 outcomes overlapped between meta-analyses of RCTs and of observational studies, and thus in total 102 unique outcomes were reported across all included systematic reviews. Characteristics of the included systematic reviews for each outcome are summarized in Tables B and C in S1 Text.

**Table 2. Summary of evidence grading for meta-analyses of observational epidemiological studies on menopausal hormone therapy and incidence of diseases and other health outcomes.**

| Evidence | Criteria | Meta-analyses of observational epidemiological studies[a] | |
|---|---|---|---|
| | | Outcomes with beneficial effects | Outcomes with harmful effects |
| Consistent | 95% CI of the mean effect excludes null value with no heterogeneity, or predictive distribution[b] contains an extreme proportion (>90%) of true effects in the direction of the mean effect | Esophageal cancer, gastric cancer, colorectal cancer, breast-cancer-specific survival[c], breast cancer overall survival[c], ovarian cancer overall survival[d], Alzheimer disease (ET), cataract, coronary heart disease, all-cause mortality | Breast cancer (EPT), endometrial cancer (ET), Alzheimer disease (EPT), venous thromboembolism, deep vein thrombosis, pulmonary embolism, cholelithiasis, asthma |
| Highly suggestive | 95% CI of the mean effect excludes null value, with heterogeneity present but predictive distribution not estimable[e], or predictive distribution contains a substantial proportion (70%–90%) of true effects in the direction of the mean effect | Glioma | Breast cancer (ET), ovarian cancer, meningioma, thyroid cancer |
| Suggestive | 95% CI of the mean effect includes null value, predictive distribution not estimable[e], and 95% CI of the most precise study[f] excludes null value | Breast cancer recurrence[d], lung cancer overall survival[g], diabetes mellitus, coronary heart disease mortality, cardiovascular disease incidence and mortality | Cutaneous melanoma (ET), endometrial cancer (EPT), systemic lupus erythematosus, Parkinson disease |
| Controversial | Predictive distribution contains a non-negligible proportion (>30%) of true effects in both the same and the opposite direction of the mean effect | Breast cancer mortality, pancreatic cancer, lung cancer | |
| Insufficient | Insufficient evidence to draw conclusions | Primary liver cancer, endometrial cancer mortality, ovarian cancer recurrence, head and neck cancer, cutaneous melanoma (EPT), osteoarthritis, dementia | |

Small-study effects existed for breast-cancer-specific survival, breast cancer overall survival, and glioma; the Egger regression test was used to examine whether smaller studies tended to show more pronounced effects than larger studies; it was applied only in meta-analyses of ≥10 studies; more information is available in S3 Text. Meta-analysis results were robust to severe or extreme publication bias for esophageal cancer, gastric cancer, colorectal cancer, breast cancer (EPT), breast cancer (ET), breast-cancer-specific survival, breast cancer overall survival, endometrial cancer (ET), ovarian cancer incidence and overall survival, coronary heart disease, venous thromboembolism, asthma, and cholelithiasis; "robust to severe or extreme publication bias" means that the meta-analysis results cannot be explained away by hypothetical publication bias that greatly exceeds empirical estimates of publication bias severity in medicine (i.e., hypothetical publication bias in which statistically significant positive effects are 4-fold more likely to be published and meta-analyzed than nonsignificant or negative effects); more information is available in S3 Text. CI, confidence interval; EPT, estrogen plus progestin therapy; ET, estrogen-alone therapy.

[a]The average effects of any menopausal hormone therapy (ET or EPT) in perimenopausal or postmenopausal women. When subgroup analysis by type of hormone therapy indicated a qualitative difference or statistically significant results were found for only 1 type of hormone therapy, results are presented separately for them. The effects refer to outcome incidence, unless otherwise indicated.

[b]The predictive distribution describes how the true effect sizes across studies are distributed around the summary average effect. Predictive distribution was estimated only in meta-analyses of ≥10 studies. More information is available in S3 Text.

[c]Use of menopausal hormone therapy before or after diagnosis of cancer.

[d]Use of menopausal hormone therapy after diagnosis of cancer.

[e]Due to a small number of studies (<10) being included in the meta-analysis.

[f]The study with the smallest standard error in each meta-analysis.

[g]Use of menopausal hormone therapy before diagnosis of cancer.

## Quality assessment of included studies

For the 7 AMSTAR 2 critical domains, 36% of the included systematic reviews established a priori a protocol for the review, 59% performed a comprehensive literature search, 34% provided a list of excluded studies with justification, 78% used a satisfactory technique for assessing the risk of bias in individual studies, 47% used the random-effects model for meta-analysis, 47% discussed the impact of risk of bias in individual studies in the interpretation of the results of the review, and 29% performed graphical or statistical tests for publication bias and discussed the likelihood and magnitude of impact of publication bias. Fig 2 presents a summary

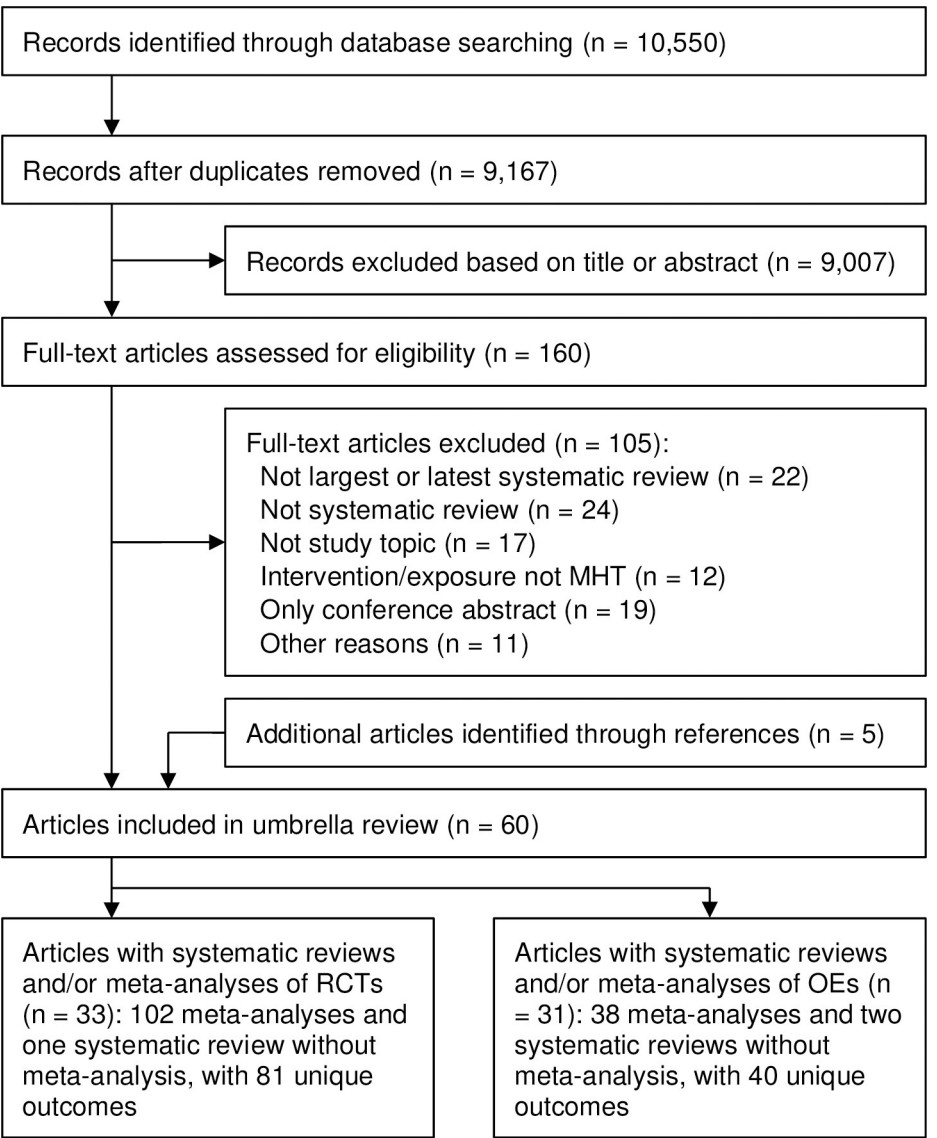

**Fig 1. Flow chart for study selection process.** MHT, menopausal hormone therapy; OE, observational epidemiological study; RCT, randomized controlled trial.

of quality assessment by outcome across all included systematic reviews. Each AMSTAR 2 domain judgment for each outcome is available in Tables D and E in S1 Text.

## Benefits and harms of MHT

In total, we included 936 individual study estimates from RCTs and 380 from observational studies (190 from case–control and 190 from cohort studies) for meta-analysis. The median number of study estimates per outcome in meta-analyses of RCTs and of observational studies was 5 (range 1–55) and 7 (range 1–71), respectively. Tables 1 and 2 summarize the evidence for all disease outcomes (both primary and secondary prevention) from meta-analyses of RCTs and of observational studies, respectively. The surrogate outcomes with consistent or highly suggestive evidence from meta-analyses of RCTs are summarized in Fig C in S1 Text.

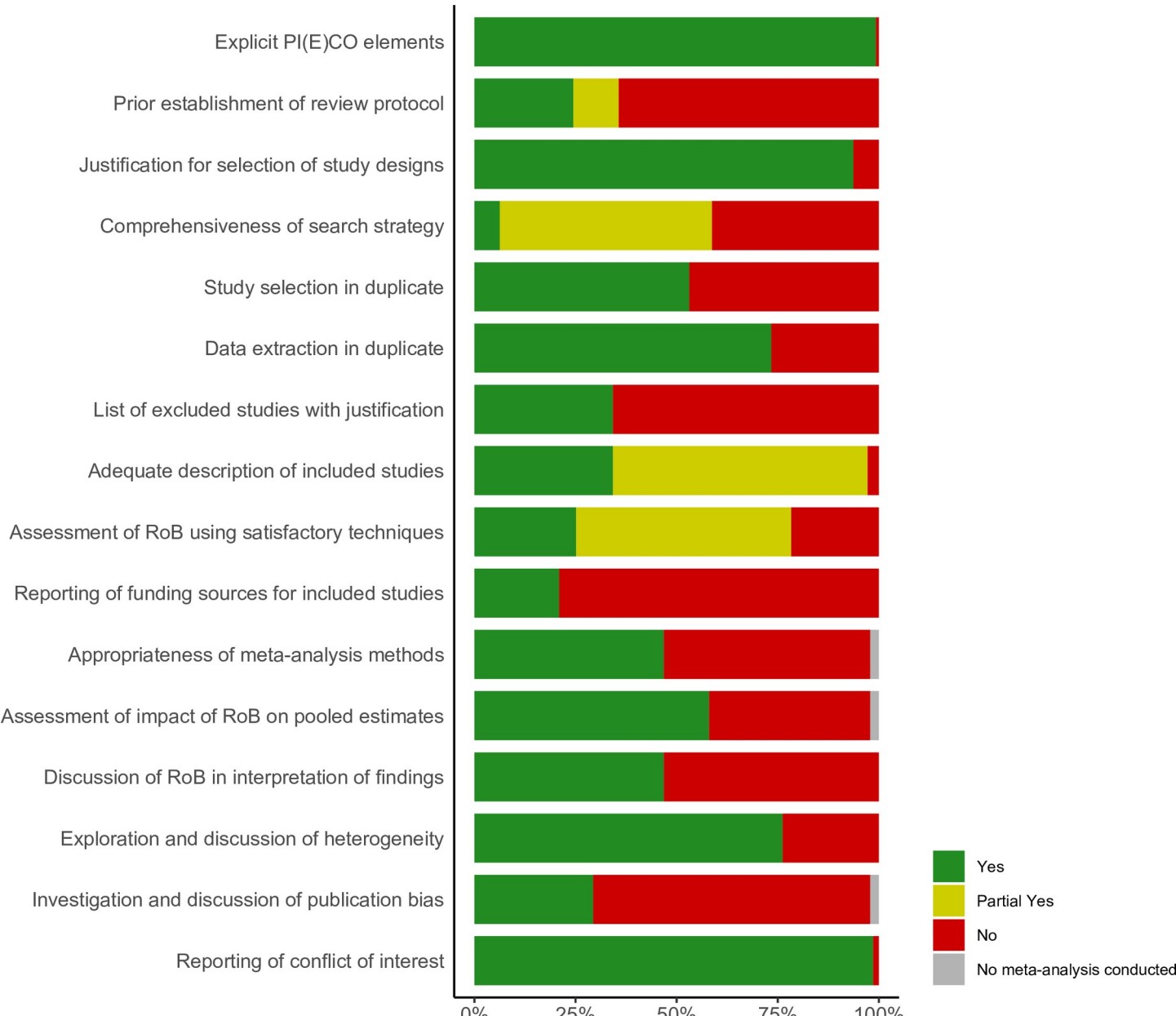

**Fig 2. Quality assessment by outcome presented as percentages across all included systematic reviews.** PI(E)CO, Population, Intervention or Exposure, Comparator, Outcome; RoB, risk of bias.

Consistent or highly suggestive evidence means that the 95% CI of the summary average effect excludes the null or the estimated predictive distribution contains a substantial proportion (≥70%) of true effects in the direction of the average effect (Tables 1 and 2). More detailed results are available in Tables F to X in S1 Text. Below we described the meta-analysis results for only outcomes with consistent or highly suggestive evidence (Figs 3 and 4, and Fig C in S1 Text). The meta-analysis results for other evidence levels, as well as small-study effects, publication bias, and sensitivity analysis for residual confounding in observational studies, are described in S1 Text.

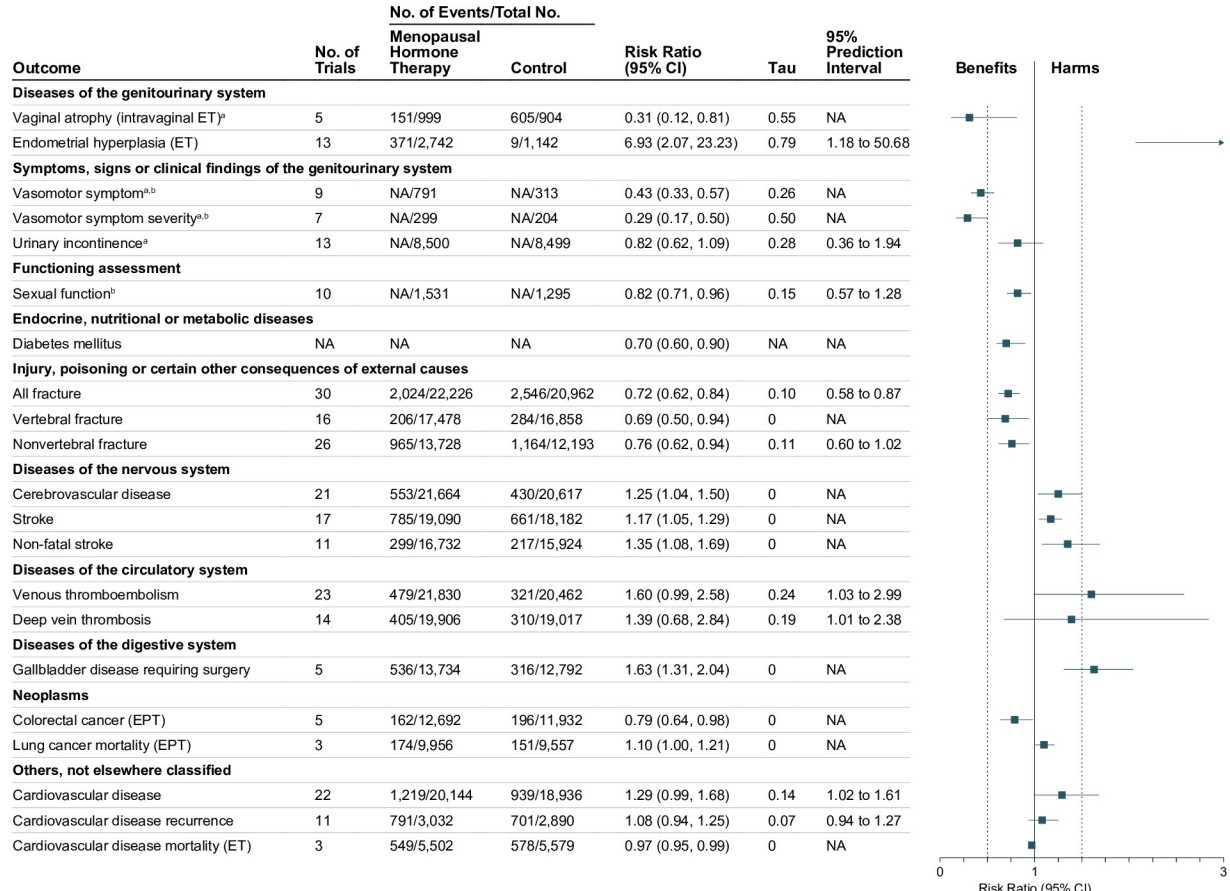

| Outcome | No. of Trials | No. of Events/Total No. | | Risk Ratio (95% CI) | Tau | 95% Prediction Interval |
|---|---|---|---|---|---|---|
| | | Menopausal Hormone Therapy | Control | | | |
| **Diseases of the genitourinary system** | | | | | | |
| Vaginal atrophy (intravaginal ET)[a] | 5 | 151/999 | 605/904 | 0.31 (0.12, 0.81) | 0.55 | NA |
| Endometrial hyperplasia (ET) | 13 | 371/2,742 | 9/1,142 | 6.93 (2.07, 23.23) | 0.79 | 1.18 to 50.68 |
| **Symptoms, signs or clinical findings of the genitourinary system** | | | | | | |
| Vasomotor symptom[a,b] | 9 | NA/791 | NA/313 | 0.43 (0.33, 0.57) | 0.26 | NA |
| Vasomotor symptom severity[a,b] | 7 | NA/299 | NA/204 | 0.29 (0.17, 0.50) | 0.50 | NA |
| Urinary incontinence[a] | 13 | NA/8,500 | NA/8,499 | 0.82 (0.62, 1.09) | 0.28 | 0.36 to 1.94 |
| **Functioning assessment** | | | | | | |
| Sexual function[b] | 10 | NA/1,531 | NA/1,295 | 0.82 (0.71, 0.96) | 0.15 | 0.57 to 1.28 |
| **Endocrine, nutritional or metabolic diseases** | | | | | | |
| Diabetes mellitus | NA | NA | NA | 0.70 (0.60, 0.90) | NA | NA |
| **Injury, poisoning or certain other consequences of external causes** | | | | | | |
| All fracture | 30 | 2,024/22,226 | 2,546/20,962 | 0.72 (0.62, 0.84) | 0.10 | 0.58 to 0.87 |
| Vertebral fracture | 16 | 206/17,478 | 284/16,858 | 0.69 (0.50, 0.94) | 0 | NA |
| Nonvertebral fracture | 26 | 965/13,728 | 1,164/12,193 | 0.76 (0.62, 0.94) | 0.11 | 0.60 to 1.02 |
| **Diseases of the nervous system** | | | | | | |
| Cerebrovascular disease | 21 | 553/21,664 | 430/20,617 | 1.25 (1.04, 1.50) | 0 | NA |
| Stroke | 17 | 785/19,090 | 661/18,182 | 1.17 (1.05, 1.29) | 0 | NA |
| Non-fatal stroke | 11 | 299/16,732 | 217/15,924 | 1.35 (1.08, 1.69) | 0 | NA |
| **Diseases of the circulatory system** | | | | | | |
| Venous thromboembolism | 23 | 479/21,830 | 321/20,462 | 1.60 (0.99, 2.58) | 0.24 | 1.03 to 2.99 |
| Deep vein thrombosis | 14 | 405/19,906 | 310/19,017 | 1.39 (0.68, 2.84) | 0.19 | 1.01 to 2.38 |
| **Diseases of the digestive system** | | | | | | |
| Gallbladder disease requiring surgery | 5 | 536/13,734 | 316/12,792 | 1.63 (1.31, 2.04) | 0 | NA |
| **Neoplasms** | | | | | | |
| Colorectal cancer (EPT) | 5 | 162/12,692 | 196/11,932 | 0.79 (0.64, 0.98) | 0 | NA |
| Lung cancer mortality (EPT) | 3 | 174/9,956 | 151/9,557 | 1.10 (1.00, 1.21) | 0 | NA |
| **Others, not elsewhere classified** | | | | | | |
| Cardiovascular disease | 22 | 1,219/20,144 | 939/18,936 | 1.29 (0.99, 1.68) | 0.14 | 1.02 to 1.61 |
| Cardiovascular disease recurrence | 11 | 791/3,032 | 701/2,890 | 1.08 (0.94, 1.25) | 0.07 | 0.94 to 1.27 |
| Cardiovascular disease mortality (ET) | 3 | 549/5,502 | 578/5,579 | 0.97 (0.95, 0.99) | 0 | NA |

**Fig 3. Consistent or highly suggestive evidence from meta-analyses of randomized controlled trials on menopausal hormone therapy and incidence of diseases and other health outcomes.** The average effects of any menopausal hormone therapy (ET or EPT) in perimenopausal or postmenopausal women, unless otherwise stated. All estimates are from our own analysis apart from diabetes mellitus. Subgroup analysis results by type of menopausal hormone therapy can be found in Tables F to K in S1 Text. The center of each square represents the summary average effect for each outcome, and the horizontal line represents the corresponding 95% CI. CI, confidence interval; EPT, estrogen plus progestin therapy; ET, estrogen-alone therapy; NA, not available or not applicable. [a]In women who already have the outcome of interest. [b]The effect measures for continuous outcomes were converted into the risk ratio scale for comparison; the results in original scale can be found in Tables F and I in S1 Text.

**Neoplasms.** In RCTs, EPT was associated with a decreased risk of colorectal cancer (5 trials, 24,624 women, RR 0.79, 95% CI 0.64 to 0.98, $p = 0.045$), but an increased risk of lung cancer mortality (3 trials, 19,513 women, RR 1.10, 95% CI 1.00 to 1.21, $p = 0.047$). In observational studies, MHT was associated with decreased risks of glioma (10 studies, 1,580,830 women, RR 0.87, 95% CI 0.72 to 1.04, $p = 0.11$, 95% PI 0.57 to 1.21), esophageal cancer (5 studies, 203,548 women, RR 0.70, 95% CI 0.60 to 0.81, $p = 0.009$), gastric cancer (6 studies, 616,630 women, RR 0.78, 95% CI 0.70 to 0.86, $p = 0.003$), and colorectal cancer (25 studies, 527,776 women, RR 0.83, 95% CI 0.77 to 0.89, $p < 0.001$, 95% PI 0.57 to 1.06). Among women with a history of breast cancer, both pre- and post-diagnosis MHT use was associated with improved breast-cancer-specific survival (11 studies, 24,753 women, RR 0.72, 95% CI 0.59 to 0.88, $p = 0.006$, 95% PI 0.48 to 0.93) and breast cancer overall survival (16 studies, 39,593 women, RR 0.82, 95% CI 0.75 to 0.89, $p < 0.001$, 95% PI 0.59 to 1.06), and among women with a history of ovarian cancer, post-diagnosis MHT use was associated with improved ovarian cancer overall survival (3 studies, 599 women, RR 0.81, 95% CI 0.71 to 0.91, $p = 0.025$). On the

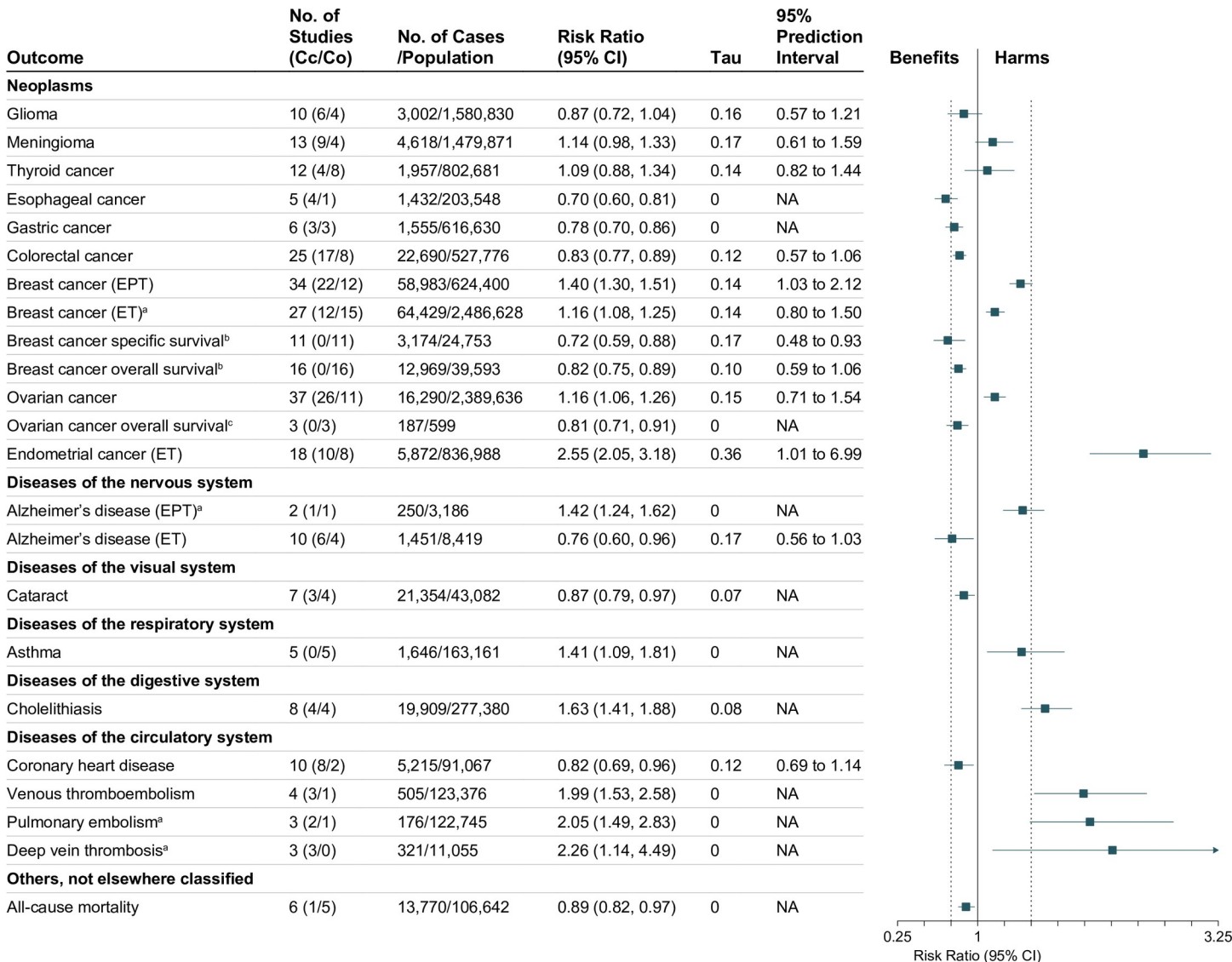

| Outcome | No. of Studies (Cc/Co) | No. of Cases /Population | Risk Ratio (95% CI) | Tau | 95% Prediction Interval | Benefits / Harms |
|---|---|---|---|---|---|---|
| **Neoplasms** | | | | | | |
| Glioma | 10 (6/4) | 3,002/1,580,830 | 0.87 (0.72, 1.04) | 0.16 | 0.57 to 1.21 | |
| Meningioma | 13 (9/4) | 4,618/1,479,871 | 1.14 (0.98, 1.33) | 0.17 | 0.61 to 1.59 | |
| Thyroid cancer | 12 (4/8) | 1,957/802,681 | 1.09 (0.88, 1.34) | 0.14 | 0.82 to 1.44 | |
| Esophageal cancer | 5 (4/1) | 1,432/203,548 | 0.70 (0.60, 0.81) | 0 | NA | |
| Gastric cancer | 6 (3/3) | 1,555/616,630 | 0.78 (0.70, 0.86) | 0 | NA | |
| Colorectal cancer | 25 (17/8) | 22,690/527,776 | 0.83 (0.77, 0.89) | 0.12 | 0.57 to 1.06 | |
| Breast cancer (EPT) | 34 (22/12) | 58,983/624,400 | 1.40 (1.30, 1.51) | 0.14 | 1.03 to 2.12 | |
| Breast cancer (ET)[a] | 27 (12/15) | 64,429/2,486,628 | 1.16 (1.08, 1.25) | 0.14 | 0.80 to 1.50 | |
| Breast cancer specific survival[b] | 11 (0/11) | 3,174/24,753 | 0.72 (0.59, 0.88) | 0.17 | 0.48 to 0.93 | |
| Breast cancer overall survival[b] | 16 (0/16) | 12,969/39,593 | 0.82 (0.75, 0.89) | 0.10 | 0.59 to 1.06 | |
| Ovarian cancer | 37 (26/11) | 16,290/2,389,636 | 1.16 (1.06, 1.26) | 0.15 | 0.71 to 1.54 | |
| Ovarian cancer overall survival[c] | 3 (0/3) | 187/599 | 0.81 (0.71, 0.91) | 0 | NA | |
| Endometrial cancer (ET) | 18 (10/8) | 5,872/836,988 | 2.55 (2.05, 3.18) | 0.36 | 1.01 to 6.99 | |
| **Diseases of the nervous system** | | | | | | |
| Alzheimer's disease (EPT)[a] | 2 (1/1) | 250/3,186 | 1.42 (1.24, 1.62) | 0 | NA | |
| Alzheimer's disease (ET) | 10 (6/4) | 1,451/8,419 | 0.76 (0.60, 0.96) | 0.17 | 0.56 to 1.03 | |
| **Diseases of the visual system** | | | | | | |
| Cataract | 7 (3/4) | 21,354/43,082 | 0.87 (0.79, 0.97) | 0.07 | NA | |
| **Diseases of the respiratory system** | | | | | | |
| Asthma | 5 (0/5) | 1,646/163,161 | 1.41 (1.09, 1.81) | 0 | NA | |
| **Diseases of the digestive system** | | | | | | |
| Cholelithiasis | 8 (4/4) | 19,909/277,380 | 1.63 (1.41, 1.88) | 0.08 | NA | |
| **Diseases of the circulatory system** | | | | | | |
| Coronary heart disease | 10 (8/2) | 5,215/91,067 | 0.82 (0.69, 0.96) | 0.12 | 0.69 to 1.14 | |
| Venous thromboembolism | 4 (3/1) | 505/123,376 | 1.99 (1.53, 2.58) | 0 | NA | |
| Pulmonary embolism[a] | 3 (2/1) | 176/122,745 | 2.05 (1.49, 2.83) | 0 | NA | |
| Deep vein thrombosis[a] | 3 (3/0) | 321/11,055 | 2.26 (1.14, 4.49) | 0 | NA | |
| **Others, not elsewhere classified** | | | | | | |
| All-cause mortality | 6 (1/5) | 13,770/106,642 | 0.89 (0.82, 0.97) | 0 | NA | |

**Fig 4. Consistent or highly suggestive evidence from meta-analyses of observational epidemiological studies on menopausal hormone therapy and incidence of diseases and other health outcomes.** The average effects of any menopausal hormone therapy (ET or EPT) in perimenopausal or postmenopausal women, unless otherwise stated. The estimates are for ever use of menopausal hormone therapy, unless otherwise stated. All estimates are from our own analysis. Subgroup analysis results by type and recency of menopausal hormone therapy use can be found in Tables L to Q in S1 Text. The center of each square represents the summary average effect for each outcome, and the horizontal line represents the corresponding 95% CI. Cc, case–control study; CI, confidence interval; Co, cohort study; EPT, estrogen plus progestin therapy; ET, estrogen-alone therapy; NA, not available or not applicable. [a]Current use of menopausal hormone therapy. [b]Use of menopausal hormone therapy before or after diagnosis of cancer. [c]Use of menopausal hormone therapy after diagnosis of cancer.

other hand, MHT was associated with increased risks of meningioma (13 studies, 1,479,871 women, RR 1.14, 95% CI 0.98 to 1.33, $p$ = 0.076, 95% PI 0.61 to 1.59), thyroid cancer (12 studies, 802,681 women, RR 1.09, 95% CI 0.88 to 1.34, $p$ = 0.36, 95% PI 0.82 to 1.44), ovarian cancer (37 studies, 2,389,636 women, RR 1.16, 95% CI 1.06 to 1.26, $p$ = 0.002, 95% PI 0.71 to 1.54), and breast cancer (71 studies, 3,331,883 women, RR 1.25, 95% CI 1.19 to 1.31, $p$ < 0.001, 95% PI 0.93 to 1.74; Table L in S1 Text). No qualitative difference in breast cancer risk between ET and EPT was found (Fig 4). In women with a uterus, ET was associated with an increased risk

of endometrial cancer (18 studies, 836,988 women, RR 2.55, 95% CI 2.05 to 3.18, $p < 0.001$, 95% PI 1.01 to 6.99).

**Diseases of the circulatory system.** In RCTs, MHT was associated with increased risks of venous thromboembolism (23 trials, 42,292 women, RR 1.60, 95% CI 0.99 to 2.58, $p = 0.052$, 95% PI 1.03 to 2.99) and deep vein thrombosis (14 trials, 38,923 women, RR 1.39, 95% CI 0.68 to 2.84, $p = 0.19$, 95% PI 1.01 to 2.38). No evidence of effect was found with regard to CHD incidence (17 trials, 39,448 women, RR 1.02, 95% CI 0.82 to 1.26, $p = 0.75$; Table F in S1 Text) and recurrence (8 trials, 5,045 women, RR 0.97, 95% CI 0.76 to 1.24, $p = 0.65$; Table I in S1 Text). In observational studies, MHT was associated with a decreased risk of CHD (10 studies, 91,067 women, RR 0.82, 95% CI 0.69 to 0.96, $p = 0.021$, 95% PI 0.69 to 1.14), but increased risks of venous thromboembolism (4 studies, 123,376 women, RR 1.99, 95% CI 1.53 to 2.58, $p = 0.006$), deep vein thrombosis (3 studies, 11,055 women, RR 2.26, 95% CI 1.14 to 4.49, $p = 0.038$), and pulmonary embolism (3 studies, 122,745 women, RR 2.05, 95% CI 1.49 to 2.83, $p = 0.017$).

**Genitourinary system.** In RCTs, MHT was associated with improved vasomotor symptoms (frequency: 9 trials, 1,104 women, RR 0.43, 95% CI 0.33 to 0.57, $p < 0.001$; severity: 7 trials, 503 women, RR 0.29, 95% CI 0.17 to 0.50, $p = 0.002$) and urinary incontinence (13 trials, 16,999 women, RR 0.82, 95% CI 0.62 to 1.09, $p = 0.15$, 95% PI 0.36 to 1.94). Intravaginal ET was associated with improved vaginal atrophy (5 trials, 1,903 women, RR 0.31, 95% CI 0.12 to 0.81, $p = 0.03$). On the other hand, oral ET was associated with an increased risk of endometrial hyperplasia (13 trials, 3,884 women, RR 6.93, 95% CI 2.07 to 23.23, $p = 0.007$, 95% PI 1.18 to 50.68).

**Functioning assessment.** In RCTs, MHT was associated with improved sexual function (measured by a composite score of arousal and sexual interest, orgasm, and pain) (10 trials, 2,826 women, RR 0.82, 95% CI 0.71 to 0.96, $p = 0.017$, 95% PI 0.57 to 1.28).

**Bone loss and fracture.** In RCTs, MHT was associated with increased bone mineral density at lumbar spine, forearm, femoral neck, and proximal femur (Fig C in S1 Text), and decreased risks of all fracture (30 trials, 43,188 women, RR 0.72, 95% CI 0.62 to 0.84, $p = 0.002$, 95% PI 0.58 to 0.87), vertebral fracture (16 trials, 34,336 women, RR 0.69, 95% CI 0.50 to 0.94, $p = 0.033$), and nonvertebral fracture (26 trials, 25,921 women, RR 0.76, 95% CI 0.62 to 0.94, $p = 0.025$, 95% PI 0.60 to 1.02).

**Diseases of the nervous system.** In RCTs, MHT was associated with increased risks of cerebrovascular disease (21 trials, 42,281 women, RR 1.25, 95% CI 1.04 to 1.50, $p = 0.03$), stroke (17 trials, 37,272 women, RR 1.17, 95% CI 1.05 to 1.29, $p = 0.027$), and nonfatal stroke (11 trials, 32,656 women, RR 1.35, 95% CI 1.08 to 1.69, $p = 0.025$). In observational studies, ET was associated with a decreased risk of Alzheimer disease (10 studies, 8,419 women, RR 0.76, 95% CI 0.60 to 0.96, $p = 0.028$, 95% PI 0.56 to 1.03), while EPT was associated with an increased risk (2 studies, 3,186 women, RR 1.42, 95% CI 1.24 to 1.62, $p = 0.02$).

**Diseases of the visual system.** In observational studies, MHT was associated with a decreased risk of cataract (7 studies, 43,082 women, RR 0.87, 95% CI 0.79 to 0.97, $p = 0.022$).

**Diseases of the respiratory system.** In observational studies, MHT was associated with an increased risk of asthma (5 studies, 163,161 women, RR 1.41, 95% CI 1.09 to 1.81, $p = 0.023$).

**Diseases of the digestive system.** In RCTs, MHT was associated with an increased risk of gallbladder disease requiring surgery (5 trials, 26,526 women, RR 1.63, 95% CI 1.31 to 2.04, $p = 0.011$). In observational studies, MHT was associated with an increased risk of cholelithiasis (8 studies, 277,380 women, RR 1.63, 95% CI 1.41 to 1.88, $p < 0.001$).

**Endocrine, nutritional, and metabolic diseases.** In RCTs, MHT was associated with lower levels of fasting glucose and fasting insulin and reduced insulin resistance in women

with and without diabetes mellitus (Fig C in S1 Text), and a decreased risk of developing diabetes mellitus (RR 0.70, 95% CI 0.60 to 0.90; Table X in S1 Text). In addition, MHT was associated with lower levels of low-density lipoprotein cholesterol (LDL-C), lipoprotein(a), and plasminogen activator inhibitor-1 (PAI-1), but higher levels of triglycerides and C-reactive protein (Fig C in S1 Text).

**Other diseases not elsewhere classified.** In RCTs, MHT was associated with increased risks of cardiovascular disease incidence (22 trials, 39,080 women, RR 1.29, 95% CI 0.99 to 1.68, $p$ = 0.056, 95% PI 1.02 to 1.61) and recurrence (11 trials, 5,922 women, RR 1.08, 95% CI 0.94 to 1.25, $p$ = 0.18, 95% PI 0.94 to 1.27), while ET was associated with a small reduction in cardiovascular mortality (3 trials, 11,081 women, RR 0.97, 95% CI 0.95 to 0.99, $p$ = 0.039). In observational studies, MHT was associated with a decreased risk of all-cause mortality (6 studies, 106,642 women, RR 0.89, 95% CI 0.82 to 0.97, $p$ = 0.029), but no evidence of effect was found in RCTs (38 trials, 47,757 women, RR 0.99, 95% CI 0.83 to 1.18, $p$ = 0.72; Table F in S1 Text).

## Discussion

### Summary of key findings

In this umbrella review, consistent or highly suggestive evidence from RCTs found that MHT was beneficial for vasomotor symptoms, vaginal atrophy (intravaginal ET), sexual function, all fracture, vertebral and nonvertebral fracture, diabetes mellitus, cardiovascular mortality (ET), and colorectal cancer (EPT), but harmful for cardiovascular disease incidence and recurrence, cerebrovascular disease, stroke, nonfatal stroke, venous thromboembolism, deep vein thrombosis, gallbladder disease requiring surgery, and lung cancer mortality (EPT). Consistent or highly suggestive evidence from observational studies found that MHT was associated with decreased risks of cataract, glioma, and esophageal, gastric, and colorectal cancer, but increased risks of venous thromboembolism, deep vein thrombosis, pulmonary embolism, cholelithiasis, asthma, meningioma, and thyroid, breast, and ovarian cancer. The effects of ET and EPT differed qualitatively for endometrial cancer, endometrial hyperplasia, and Alzheimer disease. The evidence levels for the other disease outcomes ranged from suggestive to insufficient.

### Limitations

Certain limitations need to be taken into account in the interpretation of our findings. First, umbrella reviews focus on existing systematic reviews, and therefore outcomes not assessed in a systematic review cannot be included. Second, the studies identified in the updated search in 2020 were not included in the umbrella review. This evidence analysis will be taken into account in the future when the current umbrella review is updated. Third, we were unable to take into account the quality of the individual studies included in the systematic reviews. Quality appraisal of individual studies is not the focus of umbrella reviews, as included systematic reviews are expected to perform this task. However, as discussed later, included systematic reviews are generally limited in their usefulness for accurate rating of quality of evidence. Fourth, it can be expected that the effects of MHT may vary between participants (e.g., based on age or time since menopause) and treatments (MHT type, dose, formulation, duration of use, and route of administration). Due to data unavailability, however, we were unable to address varying effects based on these prognostic factors, particularly for the commonly used MHT formulations (e.g., estradiol and micronized progesterone). Instead, we used the predictive distribution (e.g., 95% PI) to estimate the range of effects across study settings [40–42]. Thus, the random-effects average effect should be interpreted in conjunction with the

predictive distribution and the effect from the subset of studies most relevant to the patients needed to treat [112,113]. Further systematic reviews or umbrella reviews of comparative evidence (e.g., head-to-head randomized trials) on different prognostic factors are warranted.

## Breast and ovarian cancer

In observational studies, we found that both ET and EPT were associated with increased incidence of breast and ovarian cancer. Our findings are consistent with those of 2 recent individual participant meta-analyses of the worldwide epidemiological evidence [7,8]. However, among women with a history of breast or ovarian cancer, pre- or post-diagnosis MHT use was associated with improved cancer-specific or overall survival. The findings were further replicated by 2 recent prospective cohort studies [114,115]. One possible explanation for these findings could be that women who used MHT after diagnosis were likely to have used MHT before diagnosis. Therefore, despite increased incidence of breast and ovarian cancer with MHT use, MHT users with established breast and ovarian tumors may have better prognosis than nonusers [114,115]. Observational data on MHT and breast cancer mortality were controversial. Subgroup analysis by MHT type did not account for the divergent results. Data from the Million Women Study recently showed that both ET and EPT were associated with increased breast cancer mortality [116]. It is therefore difficult to interpret the discrepancy in these results, but one speculation could be that any causal effect of MHT on breast cancer survival may reasonably result in such divergent results, since mortality reflects the net effects of MHT on both incidence and survival [117]. Nevertheless, it is unclear whether the inverse association between MHT and breast cancer survival is causal or due to residual confounding or other biases. In our analysis, small-study effects were also present. Furthermore, the HABITS (Hormonal Replacement Therapy after Breast Cancer—Is It Safe?) trial [118,119], which compared MHT for menopausal symptoms with symptomatic treatment without MHT among women with a history of breast cancer, found an increased risk of breast cancer recurrence following MHT. The HABITS trial was therefore terminated early [118], along with the Stockholm trial [120,121], in which, however, no evidence of increased breast cancer recurrence with MHT was found. Current guidelines [5,11,122] suggest that, in women with a history of breast cancer, systemic MHT is generally not recommended for alleviating vasomotor symptoms, and low-dose vaginal ET may be an option to manage genitourinary symptoms after nonhormonal therapies or complementary options have been unsuccessful and after a detailed discussion of risks and benefits and review with an oncologist. In all, given the paucity of data from RCTs and the potential biases in the observational evidence, further well-designed longitudinal studies of MHT use in women with a history of breast or ovarian cancer across different settings, while controlling for different sources of bias, are warranted to assess causality.

In the WHI trials, a qualitative difference was found in the effects of ET and EPT on breast cancer incidence, with ET decreasing but EPT increasing the risk [31]. ET was also found to be associated with lower breast cancer mortality, whereas no evidence of effect was observed for EPT [31]. Several explanations were proposed to reconcile the discordance between the WHI trials and observational studies: older women in the WHI trials, confounding in observational studies, different biological mechanisms of ET and EPT, or simply the play of chance in the WHI trials [8,31]. Given the extensive amount of epidemiological evidence, replication of the findings from the WHI trials should be undertaken.

## CHD and all-cause mortality

The effects of MHT on CHD and all-cause mortality have long been debated [90,123–128]. In our analysis, observational data consistently showed up to 18% lower risk of CHD and up to

11% lower risk of all-cause mortality with ever use of MHT, while RCTs failed to support the presence of these beneficial effects. The distinction in populations of women between RCTs and observational studies—i.e., that observational studies generally included younger women who started MHT around the time of menopause—has led to the well-known "timing hypothesis" [123,124]. According to this hypothesis, MHT reduces CHD and all-cause mortality only when initiated close to the onset of menopause, but does not reduce or even increases CHD and all-cause mortality when initiated many years later [123,124]. Indeed, the age-stratification analyses from the WHI trials found that MHT reduced CHD and all-cause mortality in women aged 50–59 years, but increased CHD in older women [33,34]. Four meta-analyses of RCTs assessed the timing hypothesis by stratifying trials according to mean time since menopause or mean age of participants at baseline, showing a highly significant reduction in CHD and all-cause mortality in women aged <60 years or those <10 years from menopause, but not in older women [60,76,99,129]. The literature reviews widely adopted the WHI trials and the meta-analyses as part of the evidence supporting the beneficial effects of MHT in the younger group of women [123–126]. The results were further carried into current guidelines [5,9,11]. However, it is worth noting that all these analyses are by nature post hoc, and such subgroup results are most appropriately regarded as hypothesis generation rather than hypothesis confirmation due to multiple comparisons [10,128,130,131]. Particularly for the meta-analyses, because of issues of potential confounding and ecological fallacy [132,133], findings of such study-level analyses have to be interpreted cautiously. On the other hand, we did not find evidence that MHT caused additional events in women (mean baseline age ≥ 60 years) with established CHD.

ELITE (Early versus Late Intervention Trial with Estradiol) [134], the first trial specifically designed to test the timing hypothesis, found that oral estradiol reduced the progression of subclinical atherosclerosis (measured as carotid-artery intima–media thickness) when initiated within 6 years after menopause, but not when initiated ≥10 years after menopause. It is unclear whether this favorable effect on atherosclerosis will translate into a reduction in risk of CHD [128]. In addition, the Kronos Early Estrogen Prevention Study (KEEPS) [135], a RCT of MHT in recently menopausal women (within 3 years after menopause), found no evidence of effect of oral conjugated equine estrogens or transdermal 17β-estradiol on the progression of carotid-artery intima–media thickness or coronary artery calcium. Mendelian randomization studies have demonstrated that several plasma biomarkers (e.g., LDL-C) have a causal role in CHD [136–140]. Among these biomarkers, however, we found that the effects of MHT were mixed, with MHT reducing LDL-C, lipoprotein(a), and PAI-1, but increasing triglycerides. Overall, the available clinical data in support of the beneficial effects of MHT on CHD and all-cause mortality as well as the timing hypothesis are only suggestive. Current guidelines [5,6,10,11] suggest that for menopausal women with low cardiovascular risk and no contraindications, MHT could be considered for treatment of bothersome vasomotor symptoms and prevention of bone loss or fracture for those at elevated risk, but not for primary prevention of cardiovascular diseases. More clinical as well as biological data are needed to confirm or refute the timing hypothesis.

## Urinary incontinence

The WHI trials found that oral ET or EPT increased the risk of urinary incontinence among continent women [141]. The 2012 Cochrane review [63] included 13 RCTs, comprising 10 small studies and 3 large studies, to assess the efficacy of MHT in treating urinary incontinence. Four small studies used intravaginal ET, while the rest used oral ET or EPT. Based on post hoc subgroup analysis, the review concluded that intravaginal ET may improve

incontinence, while oral ET or EPT may worsen incontinence. Though qualitative differences between different routes of administration are generally not expected [51], this result was nevertheless carried into the guidelines without caveats [5,9]. In our analysis, small-study effects were very prominent: All 10 small studies, regardless of route of administration, pointed in the direction of beneficial effects, while the 3 large studies reported harmful effects. Taken together, current evidence shows that oral MHT increases the risk of urinary incontinence and worsens incontinence in postmenopausal women. The opposite effects of systemic and local MHT should be viewed at best as exploratory.

## Credibility of meta-analyses

We found major methodological limitations in a substantial proportion of included meta-analyses, some of which warrant further discussion. First, a fixed-effect meta-analysis model, instead of a random-effects model, was often used, even when heterogeneity was present or anticipated a priori. The fixed-effect model assumes that all studies in the meta-analysis share a common true effect size, whereas the random-effects model assumes that there is a distribution of true effect sizes [142]. By gathering studies from the published literature, the random-effects assumption is the only plausible match to the underlying effect distribution [142]. Second, when a random-effects model was used, the summary effect was often incorrectly interpreted as an estimate of a common effect, as under a fixed-effect model. Under the random-effects model, the summary effect is an estimate of the mean of a distribution of true effects across studies [112,113,142]. Third, investigators often relied heavily on statistical methods (e.g., the Egger test) to deal with publication bias. To address publication bias, there is a need to obtain a representative sample of studies on a topic (e.g., through comprehensive literature searches). Fourth, the tools used for risk of bias assessment of individual studies were generally not comprehensive, and risk of bias was often evaluated by study across outcomes rather than by outcome across studies, which limited the quality of evidence rating in existing meta-analyses.

In conclusion, MHT has a complex balance of benefits and harms on various health outcomes. Some effects differ qualitatively between ET and EPT. Decisions regarding the use of MHT should consider the full range of effects, along with patients' values and preferences. The overall quality of existing systematic reviews is moderate to poor. Clinicians should evaluate their scientific strength prior to considering applying their results in clinical practice.

## Supporting information

**S1 PRISMA Checklist.**
(DOCX)

**S1 Text. Search strategy, study characteristics, data extraction protocols, quality assessment, and analysis results (heterogeneity, publication bias, etc.).** Results: Supplementary results for umbrella review on menopausal hormone therapy and women's health. Table A: Search strategies used to retrieve papers from different databases. Table B: Characteristics of included systematic reviews and/or meta-analyses of randomized controlled trials on menopausal hormone therapy and multiple outcomes. Table C: Characteristics of included systematic reviews and/or meta-analyses of observational epidemiological studies on menopausal hormone therapy and multiple outcomes. Table D: Quality assessment of included systematic reviews and/or meta-analyses of randomized controlled trials on menopausal hormone therapy and multiple outcomes. Table E: Quality assessment of included systematic reviews and/or meta-analyses of observational epidemiological studies on menopausal hormone therapy and

multiple outcomes. Table F: Any menopausal hormone therapy for primary prevention of multiple outcomes in included systematic reviews and meta-analyses of randomized controlled trials. Table G: Estrogen-alone therapy for primary prevention of multiple outcomes in included systematic reviews and meta-analyses of randomized controlled trials. Table H: Estrogen plus progestin therapy for primary prevention of multiple outcomes in included systematic reviews and meta-analyses of randomized controlled trials. Table I: Any menopausal hormone therapy for secondary prevention of multiple outcomes in included systematic reviews and meta-analyses of randomized controlled trials. Table J: Estrogen-alone therapy for secondary prevention of multiple outcomes in included systematic reviews and meta-analyses of randomized controlled trials. Table K: Estrogen plus progestin therapy for secondary prevention of multiple outcomes in included systematic reviews and meta-analyses of randomized controlled trials. Table L: Any menopausal hormone therapy for primary prevention of multiple outcomes in included systematic reviews and meta-analyses of observational epidemiological studies. Table M: Estrogen-alone therapy for primary prevention of multiple outcomes in included systematic reviews and meta-analyses of observational epidemiological studies. Table N: Estrogen plus progestin therapy for primary prevention of multiple outcomes in included systematic reviews and meta-analyses of observational epidemiological studies. Table O: Any menopausal hormone therapy for secondary prevention of multiple outcomes in included systematic reviews and meta-analyses of observational epidemiological studies. Table P: Estrogen-alone therapy for secondary prevention of multiple outcomes in included systematic reviews and meta-analyses of observational epidemiological studies. Table Q: Estrogen plus progestin therapy for secondary prevention of multiple outcomes in included systematic reviews and meta-analyses of observational epidemiological studies. Table R: Assessment of small-study effects and publication bias: Any menopausal hormone therapy for primary prevention of multiple outcomes in included systematic reviews and meta-analyses of randomized controlled trials. Table S: Assessment of small-study effects and publication bias: Any menopausal hormone therapy for secondary prevention of multiple outcomes in included systematic reviews and meta-analyses of randomized controlled trials. Table T: Assessment of small-study effects and publication bias: Any menopausal hormone therapy for primary prevention of multiple outcomes in included systematic reviews and meta-analyses of observational epidemiological studies. Table U: Sensitivity analysis for residual confounding: Any menopausal hormone therapy for primary prevention of multiple outcomes in included systematic reviews and meta-analyses of observational epidemiological studies. Table V: Assessment of small-study effects and publication bias: Any menopausal hormone therapy for secondary prevention of multiple outcomes in included systematic reviews and meta-analyses of observational epidemiological studies. Table W: Sensitivity analysis for residual confounding: Any menopausal hormone therapy for secondary prevention of multiple outcomes in included systematic reviews and meta-analyses of observational epidemiological studies. Table X: Summary of results for outcomes with no available data for meta-analysis. Fig A: Prespecified protocol for extracting data from included systematic reviews and/or meta-analyses of randomized controlled trials on menopausal hormone therapy and multiple outcomes. Fig B: Prespecified protocol for extracting data from included systematic reviews and/or meta-analyses of observational epidemiological studies on menopausal hormone therapy and multiple outcomes. Fig C: Consistent or highly suggestive evidence from meta-analyses of randomized controlled trials on menopausal hormone therapy and multiple surrogate outcomes. Fig D: Suggestive evidence from meta-analyses of randomized controlled trials on menopausal hormone therapy and incidence of diseases and other health outcomes. Fig E: Suggestive evidence from meta-analyses of observational epidemiological studies on menopausal hormone therapy

and incidence of diseases and other health outcomes.
(PDF)

**S2 Text. Updated database search, from November 27, 2017, to December 17, 2020.**
Table A: Characteristics of systematic reviews and/or meta-analyses on menopausal hormone therapy and multiple outcomes, updated on December 17, 2020. Fig A: Flow chart for study selection process, updated on December 17, 2020.
(PDF)

**S3 Text. Statistical analysis protocol.**
(PDF)

**S4 Text. R scripts for meta-analyses of randomized controlled trials.**
(PDF)

**S5 Text. R scripts for meta-analyses of observational epidemiological studies.**
(PDF)

## Acknowledgments

We would like to thank the Biomedical Library at University of Gothenburg for retrieving the full-text articles. We also thank Sarah M. Greising (University of Minnesota, US), Mohammad Hassan Murad (Mayo Clinic, US), and James M. Whedon (Southern California University of Health Sciences, US) for providing additional data or information. We thank Muwada Bashir Awad Bashir (University of Gothenburg, Sweden) for assistance with data extraction, and Yarong Tian (University of Gothenburg, Sweden) for advice on drawing figures. They were not compensated for their contribution.

## Author Contributions

**Conceptualization:** Guo-Qiang Zhang, Siew Hwa Lee, Bright I. Nwaru.

**Data curation:** Guo-Qiang Zhang, Jin-Liang Chen, Ying Luo, Panagiotis Anagnostis, Ulugbek Nurmatov, Madar Talibov, Jing Zhang, Siew Hwa Lee, Bright I. Nwaru.

**Formal analysis:** Guo-Qiang Zhang, Jin-Liang Chen, Ying Luo, Maya B. Mathur, Panagiotis Anagnostis, Ulugbek Nurmatov, Madar Talibov, Jing Zhang, Catherine M. Hawrylowicz, Mary Ann Lumsden, Hilary Critchley, Aziz Sheikh, Bo Lundbäck, Cecilia Lässer, Hannu Kankaanranta, Siew Hwa Lee, Bright I. Nwaru.

**Funding acquisition:** Bo Lundbäck, Bright I. Nwaru.

**Investigation:** Guo-Qiang Zhang, Jin-Liang Chen, Ying Luo, Maya B. Mathur, Panagiotis Anagnostis, Ulugbek Nurmatov, Madar Talibov, Jing Zhang, Catherine M. Hawrylowicz, Mary Ann Lumsden, Hilary Critchley, Aziz Sheikh, Bo Lundbäck, Cecilia Lässer, Hannu Kankaanranta, Siew Hwa Lee, Bright I. Nwaru.

**Methodology:** Guo-Qiang Zhang, Maya B. Mathur, Bright I. Nwaru.

**Project administration:** Guo-Qiang Zhang, Bright I. Nwaru.

**Resources:** Bright I. Nwaru.

**Software:** Guo-Qiang Zhang, Maya B. Mathur.

**Supervision:** Bo Lundbäck, Cecilia Lässer, Hannu Kankaanranta, Bright I. Nwaru.

**Validation:** Guo-Qiang Zhang, Jin-Liang Chen, Ying Luo, Maya B. Mathur, Panagiotis Anagnostis, Jing Zhang, Bright I. Nwaru.

**Visualization:** Guo-Qiang Zhang, Maya B. Mathur, Hannu Kankaanranta, Bright I. Nwaru.

**Writing – original draft:** Guo-Qiang Zhang, Bright I. Nwaru.

**Writing – review & editing:** Guo-Qiang Zhang, Jin-Liang Chen, Ying Luo, Maya B. Mathur, Panagiotis Anagnostis, Ulugbek Nurmatov, Madar Talibov, Jing Zhang, Catherine M. Hawrylowicz, Mary Ann Lumsden, Hilary Critchley, Aziz Sheikh, Bo Lundbäck, Cecilia Lässer, Hannu Kankaanranta, Siew Hwa Lee, Bright I. Nwaru.

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
