## [Editor Report · Decision Letter 0]

16 Feb 2021

Dear Dr Zhang, 

Thank you for submitting your manuscript entitled "Menopausal Hormone Therapy and Women’s Health: An Umbrella Review of Systematic Reviews and Meta-Analyses of Randomized Controlled Trials and Observational Epidemiological Studies" for consideration by PLOS Medicine.

Your manuscript has now been evaluated by the PLOS Medicine editorial staff and I am writing to let you know that we would like to send your submission out for external peer review.

Please re-submit your manuscript within two working days, i.e. by February 19, 2021.

Kind regards,

Beryne Odeny

Associate Editor

PLOS Medicine

---

## [Decision Letter · Decision Letter 1]

16 Apr 2021

Dear Dr. Zhang,

Thank you very much for submitting your manuscript "Menopausal Hormone Therapy and Women’s Health: An Umbrella Review of Systematic Reviews and Meta-Analyses of Randomized Controlled Trials and Observational Epidemiological Studies" (PMEDICINE-D-21-00766R1) for consideration at PLOS Medicine. 

[LINK]

In light of these reviews, I am afraid that we will not be able to accept the manuscript for publication in the journal in its current form, but we would like to consider a revised version that addresses the reviewers' and editors' comments. Obviously we cannot make any decision about publication until we have seen the revised manuscript and your response, and we plan to seek re-review by one or more of the reviewers. 

We expect to receive your revised manuscript by May 07 2021 11:59PM. Please email us (plosmedicine@plos.org) if you have any questions or concerns.

We look forward to receiving your revised manuscript. 

Sincerely,

Beryne Odeny, 

PLOS Medicine 

plosmedicine.org

1. Please revise your title according to PLOS Medicine's style. Your title must be nondeclarative and not a question. It should begin with main concept if possible. "Effect of" should be used only if causality can be inferred, i.e., for an RCT. Please place the study design ("A randomized controlled trial," "A retrospective study," "A modelling study," etc.) in the subtitle (ie, after a colon). For example, “Menopausal Hormone Therapy and Women’s Health: An Umbrella Review”

2. Please add this statement to the manuscript's Competing Interests: "AS is an Academic Editor on PLOS Medicine's editorial board."

3. Abstract summary - At this stage, we ask that you reformat your non-technical Author Summary. The Author Summary should immediately follow the Abstract in your revised manuscript. This text is subject to editorial change and should be distinct from the scientific abstract. The summary should be accessible to a wide audience that includes both scientists and non-scientists. Please see our author guidelines for more information: https://journals.plos.org/plosmedicine/s/revising-your-manuscript#loc-author-summary.

4. Abstract:

a. Please report your abstract according to PRISMA for abstracts, following the PLOS Medicine abstract structure (Background, Methods and Findings, Conclusions) http://www.plosmedicine.org/article/info:doi/10.1371/journal.pmed.1001419 .

b. Please combine the design, data sources, eligibility criteria and results sections into one section, “Methods and findings”. 

c. Please provide synthesis / appraisal methods in the abstract.

d. Please ensure that all numbers presented in the abstract are present and identical to numbers presented in the main manuscript text.

e. Please include the actual amounts or percentages of relevant outcomes, not just risk ratios.

f. Please quantify the main results (with p values in addition to 95% CI).

g. In the last sentence of the Abstract Methods and Findings section, please describe the main limitation(s) of the study's methodology.

5. Please add the following statement, or similar, to the Methods: "This study is reported as per the Preferred Reporting Items for Systematic Reviews and Meta-Analyses (PRISMA) guideline (SChecklist)."

6. Thank you for providing your PRISMA checklist. Please replace the page numbers with paragraph numbers per section (e.g. "Methods, paragraph 1"), since the page numbers of the final published paper may be different from the page numbers in the current manuscript.

7. Please include p-values in your results section and tables

8. For your figures, please do the following:

a. Please indicate in the figure caption the meaning of the bars and whiskers

b. Please provide titles and legends for all figures (including those in Supporting Information files)

9. Please refer to high income countries rather than "Western" countries. 

10. Please reformat Ref #6 ,10, 11, 17, 24, 43, 44, 47,52, 55, 60, 61, 63, 65, 66, 69. These should have 6 names listed before “et al.” Please use the "Vancouver" style for reference formatting, and see our website for other reference guidelines https://journals.plos.org/plosmedicine/s/submission-guidelines#loc-references.

Comments from the reviewers:

Reviewer #1: The authors have performed a comprehensive 'review of reviews' on the topic of menopausal hormone therapy. The data are of interest and the authors have been appropriately circumspect in their conclusions and, for the most part, in acknowledging the caveats inherent in such an analysis. However, there are some issues with the organization of the manuscript and the practicality of some of the findings that merit further attention and revision for clarity.

The presentation is confusing. It would be better organized under organ system with both beneficial and harmful effects discussed under the same subheading. Observational studies and RCTs should also be described for each subheading and stated separately. As it currently reads, the effects of hormones are both beneficial and harmful for Alzheimer's Disease, for example. The reader must therefore toggle back to the 'benefits' section to appreciate that the authors are probably indicating that combined EPT (page 15) is harmful but ET is beneficial. 

Page 14: Under Neoplasms, it should be specified that women with a uterus taking ET alone had a higher risk of endometrial cancer. If this is not the case based on the umbrella data, then discussion needs to include the fact that more recent regimens of EPT have been fairly carefully studied and several of them significantly reduce the risk of endometrial cancer compared to background risk with not therapy. This latter point is reflected in Table 1 but not in the text.

In the discussion on page 18, the authors describe an overall beneficial effect of hormone therapy on women with pre-existing breast cancer. This is very concerning, as there is some very clear evidence of publication bias in this area, with at least one major trial being stopped early for harm (https://www.ncbi.nlm.nih.gov/pmc/articles/PMC338128/). It is simply not known if other studies that have not been published had similar findings. It is extremely important to point out this caveat, as clinicians often use data from meta analyses to inform their prescribing. Breast cancer is currently an absolute contraindication to hormone therapy in the United States, and this reviewer does not know if that is also true in other countries. The author is also directed towards the recent publication by Chlebowski, et al, containing some of the most comprehensive follow-up of women with breast cancer from the WHI trial (https://jamanetwork.com/journals/jama/fullarticle/2768806), which demonstrated a neutral effect on overall mortality with relatively short term use of hormones. Moreover, the authors are further directed to he 2010 Chlebowski publication, which contained more ominous data on breast cancer and mortality from a shorter follow-up period (https://jamanetwork.com/journals/jama/fullarticle/186747). Because these data speak to the possible earlier appearance of cancers and higher mortality in the earlier years of follow-up, it is especially clinically relevant to advise caution in interpretation of the authors' umbrella analysis.

Confusion is further exacerbated by the use of the term MHT, which could mean either EPT or ET. MHT does not appear to be defined and since it adds to the confusion of this relatively complex manuscript, suggest that ET+EPT be used when referring to both types of hormone therapy and that MHT be dropped.

It would be helpful to organize the presentation of results in terms of relative importance (for example, breast cancer might be most appropriate to discuss first, as it likely has the largest effect on morbidity and mortality). At the very least, the ordering of the results should parallel the ordering of the discussion. 

On pages 18-19, the authors are to be congratulated for a concise and accurate summary of the data to date on hormone therapy and cardiovascular disease. Further on, on page 19, the authors describe the ELITE trial and its potentially beneficial findings on vascular effects of hormones in recently menopausal women. The KEEPS Trial, which was a multicenter clinical trial and found no such benefit, should also be cited for balance (https://pubmed.ncbi.nlm.nih.gov/25069991/).

Why does Figure 3A NOT include breast cancer under Neoplasms?

Reviewer #2: The authors have conducted a thorough review of the literature. 

The data analysis section mentions that data were grouped by primary or secondary prevention, but the summary of results does not make this distinction. Please discuss any relevant differences (or similarities).

The introduction mentions study quality as a rationale for conducting the study, and Figure 2 displays relevant results, but the results regarding this are not summarized or discussed in the text.

It would be useful to indicate whether results from RCTs and observational studies tended to be consistent. 

The introduction and the discussion mention the timing hypothesis, but results from the analysis do not address this directly - please clarify whether this was not possible given the nature of the data, or whether additional analyses could be done to address this question. 

Minor comments:

p. 10: please provide a brief description of Egger's regression.

p. 12: please indicate in the text what is meant by "consistent or highly suggestive evidence"

p. 13: regarding breast cancer survival, it's not clear whether this is associated with pre-diagnosis MHT use or post-diagnosis MHT use (or both)

p. 15: what is meant by "no qualitative difference" regarding breast cancer risk comparing ET and EPT? 

pp. 20-21: please explain what is meant by "the random-effects assumption is the only plausible match to the underlying effect distribution." Also, it seems that this would be an analytic approach rather than an assumption. 

Reviewer #3: This study conducted an umbrella review of system reviews and meta analyses examining the effects of MHT on various health outcomes of women. In general, I think this is a study of good quality and of importance to the field. The results are very informative and useful to clinical guidelines on MHT. The supplementary materials are well written and provide good documentation of methods and detailed results of the study. Here are my major feedback/comments on the manuscript. Attached please find the commented manuscript with more detailed feedback/comments. 

1. Please provide line numbers of the manuscript in the future.

2. Page 7: "…Often, these guidelines incorporated systematic reviews and meta-analyses as a key evidence support for the recommendations, but with little attention to their quality and scientific validity…"

Is there any evidence of such claim? 

3. Page 8: in the "Literature Search and Eligibility Criteria", I am wondering why PubMed was not included as it is a major database for medical literature.

4. Page 9: "… We performed an updated search on December 17, 2020 for studies published since November 27, 2017…" 

I don't understand why studies from Nov 2017 to Dec 2020 were not included in this umbrella analysis. Why the original search was only from inception to 2017? I see the details of the updated search in Sup 2. However, I still don't understand why not including the latest SR/MA in the analyses. 

5. Page 10: "..A subgroup analysis by MHT type was conducted to assess whether the effect varied qualitatively between ET and EPT.."

I figured out what you mean by qualitative difference latter in the paper. But it would be helpful if you could explain what are the qualitative differences of effects between ET and EPT here. 

Also, if you conducted subgroup analysis by MHT type, I think it's more common to examine quantitative differences between the effects by therapy type.

6. Page 11: in the "Results" section, I did not see the results of the quality of studies in these SR/MA. Even some descriptions/summaries could be very useful. 

7. Page 15: when you show the RR and its 95% CI, it's also useful to show the number of studies these RR estimates are based (as you did for breast cancer).

8. Page 20: "…First, a fixed-effect meta-analysis model, instead of a random-effects model, was often used even when heterogeneity was present or anticipated a priori. By gathering studies from the published literature, the random-effects assumption is the only plausible match to the underlying effect distribution…" 

This is not true. Although the distribution of the true effect is usually assumed to be normal which is in line with the assumption of random-effect model. However, fixed-effect model is often preferred over random-effect model to adjust for bias due to unobserved covariates (e.g. different unobserved characteristics of the studies), especially when there is enough sample from each study to estimate the fixed effects of each study. I think that is the main reason why you found many studies using the fixed-effect model rather than random effect model.

9. Page 21: "Second,…,Under the random-effects model, the summary effect is an estimate of the mean of a distribution of true effects across studies…"

This is not accurate either. The interpretation of estimate from random-effect model should be the mean of the distribution of the true effect conditional on the covariates (if any) in the model and on the random effects of the studies (i.e., the RE is treated as a covariate in the model). 

So, the effect from a random-effect model is still conditional effect rather than marginal effect.

10. Page 21: "..Third, investigators often relied heavily on statistical methods (e.g., Egger's test) to deal with publication bias. To address publication bias, there is need to obtain a representative sample of studies on a topic (e.g., through comprehensive literature searches)…"

Egger's regression, along with the funnel plot, is often used to detect publication bias. To deal with the publication bias, finding studies with representative sample is ideal but may not be realistic. One can practically use the "trim-and-fill" method to alleviate the effect of publication bias. 

Reference:

Shi, L., & Lin, L. (2019). The trim-and-fill method for publication bias: Practical guidelines and recommendations based on a large database of meta-analyses. Medicine, 98(23). https://doi.org/10.1097/MD.0000000000015987

Xiaochen Dai

Reviewer #4: This umbrella review evaluates the risks and benefits of menopausal hormone therapy (MHT) in women by evaluating the systemic reviews and meta analyses available for many outcomes. This is an important topic to help guide menopause clinical practice. The attempt to review all this data is applauded. The manuscript is clearly written and easy to read. However, I suggest some significant modifications to the manuscript. 

Overall, it is becoming increasingly clear that the type (ET vs EPT), route (transdermal, oral), formulation (oral conjugated equine estrogen, estradiol, MPA, micronized progesterone), dose and duration of MHT impacts its outcomes differently. This needs to be discussed in greater detail in the manuscript and an attempt to review and present findings by these characteristics should be made for each outcome (more than a line in the limitations). Page 14/15 for neoplasms is a good example. For example, much of the data on risk of meningioma is based on older data using oral higher dose estrogens and therefore doesn't provide guidance on the use of transdermal or lower dose MHT and their relationship with meningioma risk. Similarly, breast cancer risk may be related to the type, route and duration of MHT (progesterone possibly less risk than MPA, etc..) so discussion of these different relationships should be included. I recognize that this level of detail may be difficult to obtain in an umbrella review, but a discussion about these factors should accompany the results and discussion. 

In clinical practice, many are now using estradiol and micronized progesterone so to provide the most clinical utility, at a minimum the results could display findings based on these formulations. 

Additionally, results for certain outcomes differ based on the timing of initiation of MHT (age 50 - 59, 60 - 69, 70-79). Specifically, these outcomes including those related to diseases of the circulatory system as well as dementia. Therefore, in my opinion, results for these outcomes should be displayed by age of initiation (this would impact results on page 13,14 and related tables and figures). These post hoc analyses have provided scientific theory to guide more recent RCTs including ELITE and KEEPS (the latter is not discussed in the manuscript but should). Current guidelines (such as NAMS) have incorporated these findings to demonstrate the low cardiovascular risk in women < age 60 (without known disease) who are symptomatic with vasomotor symptoms and considering initiating MHT for quality of life relieving therapy. To my knowledge, guidelines have not recommended initiation of MHT for cardiovascular benefit or primary prevention at least in the U.S. (in part likely related to the concerns discussed on page 19). On page 19, it would be of benefit to rephrase the sentence starting with "current guidelines and reviews…" since those two things are different. There is a difference between recommending MHT initiation for VMS and reassuring women that CVD risk is low, vs initiating MHT for cardiovascular benefit. I fear if this difference is not made clearer in the discussion than clinicians may misinterpret the findings and withhold quality of life benefiting therapy for symptomatic menopausal women. 

A few additional specific points: 

P 4 - Consider including an additional position statement or guideline in the second bullet point to highlight examples were recommendations are not consistent 

P 4 - For the second bullet of the 'what this study adds' I would recommend considering changing the word 'tentative' - perhaps to 'would benefit from further attention' or something similar. The end of the sentence needs to clarify that the authors are suggesting the clinical guidelines be re-examined only for this particular outcome because as it reads now some may believe they are suggesting all outcomes need to be re-evaluated. 

P 4 - The last bullet in the section 'what this study adds' would benefit from modification. The first part of the sentence and the last part of the sentence run counter to each other. I would recommend considering changing the word 'credibility' to 'scientific strength' or something similar. I would think the requirement to assess credibility should fall to the researchers participating in the systemic reviews/research, and the peer reviewers and journals publishing them, as well as the societies that are using them to guide position statements. It is the duty of clinicians to use the highest strength EBM to guide their clinical decision making when that EBM exists. (This should be adjusted in the conclusion as well. You may simply delete the last sentence of your conclusions.) 

P 14 - Genitourinary syndrome: findings only reported for ET and increased risk of endometrial hyperplasia. Information about progestogens to reduce risk of endometrial hyperplasia should be presented here as well. 

The main goal of treatment for symptomatic menopausal women (vasomotor symptoms, GSM) is to relieve their menopausal symptoms. Your umbrella review found good evidence that MHT is effective at relieving these symptoms. Without acknowledging the limits of the data availability (on dose, formulations, route, age of initiation) and quality for other outcomes at the start of the discussion clinicians may misinterpret your findings and not accurately "consider the full range of effects, along with patient's values and preferences" as you state in your conclusion.

[LINK]

---

## [Decision Letter · Decision Letter 2]

3 Jun 2021

Dear Dr. Zhang,

Thank you very much for re-submitting your manuscript "Menopausal hormone therapy and women’s health: An umbrella review" (PMEDICINE-D-21-00766R2) for review by PLOS Medicine.

I have discussed the paper with my colleagues and the academic editor and it was also seen again by four reviewers. I am pleased to say that provided the remaining editorial and production issues are dealt with we are planning to accept the paper for publication in the journal.

[LINK]

We look forward to receiving the revised manuscript by Jun 10 2021 11:59PM.   

Sincerely,

Beryne Odeny, 

Associate Editor 

PLOS Medicine

plosmedicine.org

Requests from Editors:

Thank you for addressing editorial concerns.

Comments from Reviewers:

Reviewer #1: The authors have responded appropriately to the concerns raised by the initial review.

Reviewer #2: Previous concerns have been addressed.

Reviewer #3: Thanks for answering the questions and revising the paper accordingly. I think the authors have properly revised the paper and it is now satisfactory.

I just have two more comments: 

1. I now understand why you did not include SR/MA published after 2017, but I think it's better if you could include that as a limitation of the study in the limitation section.

2. I agree that the model is "collapsible" if identity link function is used. However, many systematic reviews report RR/ORs, which usually requires log transformation in the meta analysis. So in general, you can only interpret the coefficients as conditional effects. 

Reviewer #4: Thank you for attempting to address all the concerns I raised in my comments. 

A few points:

- Line 638 - 642 is incorrect. Current guidelines do not recommend MHT for vasomotor symptoms in breast cancer survivors. There is guidance regarding low dose vaginal estrogen in this population. This line needs to be updated and it needs to be made clear the difference in recommendations between systemic HT and low dose vaginal estrogen. 

- I understand you can't include information about estradiol and micronized progesterone in detail in the paper due to lack of data, but you should at least point that out since clinicians may be reviewing this umbrella review and should know that that data is not specifically about more common formulations.

- Line 162 page 8 is not fully correct. Current practice guidelines say that for women less than age 60 or within 10 years of their LMP that the benefits greatly outweigh the risks for symptomatic women, and DO Not recommend MHT for prevention of CHD or all cause mortality. Instead of the conclusion you have here, I would summarize your findings as saying that more data is needed to evaluate the timing hypothesis (not re examine the guidelines which are appropriately supporting treatment for low risk symptomatic women). 

Line 510 - why is information on VMS included in the GU system paragraph?

LIne 520 - you need to specify what parameters of sexual function improved (desire, pain, orgasm)?

[LINK]

---

## [Decision Letter · Decision Letter 3]

12 Jul 2021

Dear Dr Zhang, 

On behalf of my colleagues and the Academic Editor, Dr. Jenny E Myers, I am pleased to inform you that we have agreed to publish your manuscript "Menopausal hormone therapy and women’s health: An umbrella review" (PMEDICINE-D-21-00766R3) in PLOS Medicine.

PRESS

Sincerely, 

Beryne Odeny 

Associate Editor 

PLOS Medicine